# IBIL Measurement and Optical Simulation of the D_I_ Center in 4H-SiC

**DOI:** 10.3390/ma16072935

**Published:** 2023-04-06

**Authors:** Wenli Jiang, Wei Cheng, Menglin Qiu, Shuai Wu, Xiao Ouyang, Lin Chen, Pan Pang, Minju Ying, Bin Liao

**Affiliations:** 1Key Laboratory of Beam Technology of Ministry of Education, College of Nuclear Science and Technology, Beijing Normal University, Beijing 100875, Chinamjying@bnu.edu.cn (M.Y.);; 2Ningbo Institute of Materials Technology & Engineering, Chinese Academy of Sciences, Ningbo 315201, China; 3Institute of Radiation Technology, Beijing Academy of Science and Technology, Beijing 100875, China

**Keywords:** IBIL, in-situ measurement, luminescence, D_I_ defect, ab initio

## Abstract

In this paper, D_I_ defects are studied via experiments and calculations. The 2 MeV H^+^ is used to carry on an ion-beam-induced luminescence (IBIL) experiment to measure the in-situ luminescence of untreated and annealed 4H-SiC at 100 K. The results show that the luminescence intensity decreases rapidly with increasing H^+^ fluence, which means the losses of optical defect centers. In addition, the evident peak at 597 nm (2.07 eV) is the characteristic peak of 4H-SiC, and the weak peak between 400 nm and 450 nm is attributed to the D_I_ optical center. Moreover, the first-principles calculation of 4H-SiC is adopted to discuss the origin of D_I_ defects. The optical transition of the defect Si_C_(C_Si_)_2_ from q = 0 to q = 1 is considered the experimental value of the D_I_ defect center.

## 1. Introduction

4H-SiC has drawn much attention due to its excellent behavior in high-power electrical devices and optical properties at low temperatures, which makes 4H-SiC widely applicable in extreme conditions. A series of experiments and simulations were studied to improve the resistance of 4H-SiC devices to high-pressure and high-intensity irradiation. The key step in making silicon carbide devices is to add dopant elements, which includes adding impurities during the growth process or ion implantation at a specific zone. Moreover, sequential annealing activates dopant elements and diffuses them homogeneously. A natural SiC defect called the D_I_ defect remains during the process, which may influence the properties of SiC.

The most remarkable property of D_I_ defects is that they can remain stable at 1700 °C and appear after various ions irradiation [1,2,3,4,5]. D_I_ defects are also found in SiC layers grown on SiC substrates [6]. The optical center of D_I_ behaves differently in SiC polytypes. There are three emission peaks at 2.625, 2.600, and 2.570 eV for 6H-SiC and two emission peaks at 2.901 eV and 2.876 eV for 4H-SiC in the photoluminescence (PL) spectrum measured at an ultralow temperature [6,7,8]. For 3C–SiC, its heterogeneous epitaxial layers on the Si substrate have an emission peak at 1.972 eV [9]. Different measurements have been used to study D_I_ defects in SiC. In addition to PL measurements, cathodoluminescence (CL), electron spin resonance (ESR), and deep-level transient spectroscopy (DLTS) have also been used to investigate the optical properties of D_I_ defects in 4H-SiC [5,6,8,10]. Compared with these measurement methods, the IBIL technique can obtain the in-situ measurement spectrum, which allows us to investigate the defect evolution with H^+^ fluence. In contrast to PL using a laser beam, IBIL uses an ion beam as the irradiation source, which can excite more electrons, and the detection limit can reach parts per million (ppm). The induced ions deposit energy through radiative recombination, in which photons are released that can be captured by detectors, and nonradiative recombination, which transfers energy to the lattice through electroacoustic coupling. IBIL can excite almost the full band energy distribution. Qiu used IBIL to measure 6H-SiC with a 2 MeV H^+^ fluence at 100 K and found that the D_I_ optical center lies between 400 and 500 nm, and the D_I_ optical center disappears as the temperature continues to increase [11]. T. Egilsson used PL to study D_I_ by measuring electron-irradiated 4H-SiC at 2 K and found that the D_I_ exciton lies at 2.901 eV [10]. After fast neutron irradiation and annealing, PL measurement at 80 K also found the D_I_ peak [1].

The structure of D_I_ defects has been widely discussed, but it is still uncertain. L. Storasta predicted that D_I_ has a pseudodonor nature and is correlated with a hole trap at Ev + 0.34 eV by the pseudodonor model in 4H-SiC [10]. A. Fissel studied D_I_ defects with PL measurements at low temperatures and thought that carbon dangling bonds were responsible for the D_I_ defects [6]. T. A. G. Eberlein proposed that nearby carbon and silicon antisite pairs have the same properties as the defect centers that generate alphabet series luminescent lines, and he linked the most stable antisite pairs to the D_I_ optical center according to the (+/0) transition level of Li lines [12]. A. Gali also studied the properties of antisite pairs in 3C-SiC and considered them to be in agreement with the isolated antisite pair as a promising model for D_I_ centers [3]. However, the thermal stability of antisite pairs was studied by atomic-level simulations, and the correlation between the D_I_ and antisite pair was denied because the latter cannot remain stable above 2000 °C, similar to the former [13]. E. Rauls studied D_I_ luminescence in 6H-SiC and denied the antisite pair because the calculated gap modes cannot match the experimental result [14]. At the same time, he proposed Si_C_(C_Si_)_2_ as the origin of the D_I_ optical center, and the production process of Si_C_(C_Si_)_2_ was V_C_Si_C_(C_Si_)_2_ → V_C_ + Si_C_(C_Si_)_2_. The local vibration mode of the Si_C_(C_Si_)_2_ complex was also calculated [15], and the result agreed well with the characteristic phonon-assisted structure of the photoluminescence of D_I_.

There has always been controversy over the structure of the D_I_ optical center. In this paper, we attempt to simulate the luminescence related to defects by first-principles calculations derived from the defect formation energy optical transition. We give a reasonable explanation of the D_I_ defect and explain D_I_ luminescence based on the experimental and calculation results. In addition, we performed a series of IBIL experiments with 2 MeV H^+^ ions to measure the D_I_ center luminescence of 4H-SiC.

## 2. Experiment and Computational Method

The 4H-SiC samples (N-type) used in the experiment are provided by TanKeBlue company. Data from the product instruction manual show that the 4H-SiC (0001) wafer has a diameter of 10 cm, a thickness of 350 μm ± 15 μm, and an N impurity concentration of 10^19^ cm^−3^. The density of the sample was 3.21 g/cm^3^. The wafer was cut into a square size of 1 mm × 1 mm. According to its reference, the band gap of 4H-SiC used in the experiments is 3.23 eV. To measure the sample phase, the X-ray diffraction (XRD) measurement was conducted with Cu K_α_ radiation at room temperature (the bandwidth is 0.017°, modulation is 0.088°). The result presented in Figure 1 shows diffraction peaks at 35.7° that are associated with the (0004) Bragg reflection, which has been measured in previous 4H-SiC XRD analyses [16,17,18]. Data from the Inorganic Crystal Structure Database (ICSD) is also used to testify to the measurement result [19]. The calculations of 4H-SiC in the paper are all based on the XRD results.

IBIL was used to measure 4H-SiC using a 2 MeV H^+^ ion beam with a diameter of 6.7 mm at 100 K at the GIC4117 2 × 1.7 MV tandem accelerator at Beijing Normal University. The current of H^+^ fluence is 16.6 nA during the whole process. The crystallization temperature of amorphous 4H-SiC is approximately 875 °C [20]. Thus, the 4H-SiC samples were annealed at 900 °C in air to improve crystalline quality before the IBIL experiment, and the annealing heating time and holding time were 150 min and 20 min, respectively. All IBIL experiments were performed in a vacuum environment.

Details of the IBIL setup used in the experiments are described in Reference [21], and the schematic diagram of the IBIL experimental setup is shown in Figure 2. In the experiment, the 4H-SiC sample is irradiated by ion beams, and photons are generated during the process. The photons are transmitted through the optical tube and fiber and collected using the Ocean Optics QE-PRO spectrometer, which allows wavelengths from 200 nm to 1000 nm. A Rutherford backscattering (RBS) dataset is collected synchronously with the corresponding IBIL spectrum to revise the fluctuation of real-time flux in the data collection system. The integral time of collecting spectrum in the experiment is set as 0.5 s. Our system can also achieve precise temperature control in the range from −196 to 600 °C, and the temperature stability in this system can remain within ±1 °C.

Density functional theory (DFT) and TD-DFT/CASTEP computations with the Perdew–Burkee–Ernzerhof (PBE) function are used to simulate the optical transition characteristics of Si_C_(C_Si_)_2_ defects in 4H-SiC. There are two steps in the calculation. First, we create a single crystal cell of 4H-SiC, as shown in Figure 3a, the size of which is set as a = b = 3.078 Å and c = 10.01 Å. Then, we optimize the created structure. Second, we expand the optimized structure to a supercell of 4 × 4 × 1, including 128 cells, and then we set the Si_C_(C_Si_)_2_ defect in the expanded structure. The positions of the defect atoms are set, as shown in Figure 3b. The energy cutoff is set as 440 eV for the plane wave, and the convergence criteria are as follows: the maximum energy change is 5 × 10^−7^ eV/atom, and the maximum force between atoms is 0.01 eV/Å. We calculate the band structure of the defect at different charge states based on these settings.

## 3. Results and Discussion

### 3.1. SRIM Simulation and the IBIL Measurement Result

In this paper, the Stopping and Range of Ions in Matter (SRIM) computations are used to simulate 2 MeV H^+^ irradiating 4H-SiC. The energy transfer of 2 MeV H^+^ ions inducing 4H-SiC is shown in Figure 4, and the nuclear and electronic stopping power behave differently during the process. From the calculation results, we can deduce that electronic stopping is the main way of losing energy, which accounts for more than 99%. The inducing depth is approximately 33 μm. Most energy is lost at a depth between 25 and 37 μm from the surface.

The high temperature will promote the decomposition of excitons, which would greatly reduce the luminescence intensity and the nonradiative recombination in 4H-SiC. Thus the thermal expansion of the luminescent bands is reduced, which makes it possible to observe fine structures. Luminescence experiments need to be performed at low temperatures, such as in liquid helium or liquid nitrogen. In addition, the radiation efficiency is improved and the luminescence intensity can be increased at low temperatures. These conditions are very important for studying shallow defects, donor–acceptor pair (DAP) luminescence, and exciton luminescence. Thus, the IBIL experiment was set with 2 MeV H^+^ at 100 K to obtain a better luminescence effect. In the IBIL measurement of untreated samples, D_I_ defects were not found. Thus, we also performed an additional experiment under the conditions of 900 °C annealed 4H-SiC. The normalized spectra of the annealed and untreated sample are compared in Figure 5a,b with the same fluence of 1.45 × 10^11^ ions/cm^2^ measured at 100 K. There is a small evident peak lying between 400 nm and 500 nm for the annealed sample, which we attribute to the D_I_ optical center according to the references [6,8,10,22]. In our previous work, we measured 6H-SiC with IBIL at 100 K [11], and a small peak between 400 and 500 nm was also measured, which is considered the D_I_ optical center. The peak was observed to appear at 800 °C and survive at 1700 °C [4,5]. The energy value of the D_I_ center is slightly larger than the experimental value in the PL measurement because of the stretching of the characteristic peak. The position of the characteristic luminescence peak does not exhibit any shift, which means that annealing has no effect on the corresponding defect energy, and the D_I_ center only appears in the annealed sample measurement, which indicates that the annealing effect increases the concentration of D_I_ defects. The concentration of the D_I_ center increases because of the improvement in layer quality [6].

The in-situ spectra of annealed 4H-SiC from 1.14 × 10^11^ to 5.18 × 10^12^ ions/cm^2^ are shown in Figure 6. The luminescence intensity drops quickly because continuous ion implantation makes the optically active centers gather into small defect clusters, and the clusters gradually aggregate into larger clusters or transform, which leads to a decrease in luminescence intensity. M. Backman also studied the simulation of incoming heavy ions in 3C-SiC at room temperature with the MD code PARCAS and believed many vacancy and interstitial defects are annihilated along the ion path due to the high temperature. Moreover, the ion incidence results in more defects because of the higher temperature in the center of the incident ion path. Hence, the change in luminescence intensity is a competitive result of the two effects [23].

A spectrum obtained at 100 K with the fluence of 1.45 × 10^11^ ions/cm^2^ for the annealed sample is shown in Figure 7. There is an evident wide band, the center of which lies at 2.06 eV (597 nm). It belongs to the yellow–green luminous zone and can be attributed to the donor–acceptor pair (DAP) [24,25]. The peak is obviously asymmetrical due to the asymmetrical distribution of the vacancies of carbon and its extended defects. It can be Gaussian decomposed into two peaks at 2.28 eV (543 nm, the full width at half maximum (FWHM) is 0.402 eV, green band) and 1.98 eV (626 nm, the FWHM is 0.488 eV, orange-red band) as shown in Figure 7. The same asymmetrical peak was also found in the 4H-SiC homoepitaxial layer grown by CVD [26]. Our measurement result is also consistent with M. Malo’s measurement [27], where IBIL was used to measure reaction bonded (RB) SiC and CVD SiC with 1 MeV H^+^ ions, and he obtained the IBIL spectrum with a broad band emission and the peak lying at approximately 597 nm. The peak at 2.95 eV (420 nm), which is attributed to the D_I_ defect, is also analyzed, and the position and FWHM of the fitted Gaussian peaks at the 2.95 eV band did not depend on fluence from 1.14 × 10^11^ to 1.3 × 10^13^ ions/cm^2^ and were 2.95–2.957 eV and 0.0688–0.0728 eV, respectively.

The evolution of the 420 nm, 543 nm, 626 nm, and 597 nm luminescence centers with the continuing injection of H^+^ fluence is shown in Figure 8. The luminous intensity of 597 nm and 420 nm decreases at almost the same rates because of the decrease in luminescence centers, and the luminescence at 597 nm lasts the whole irradiation process until the dose reaches 6.5 × 10^13^ ions/cm^2^. The luminescence peak at 420 nm nearly disappears when the fluence accumulates to 1.3 × 10^13^ ions/cm^2^.

IBIL experiments of annealed 4H-SiC at different temperatures are also performed to study the temperature effect on the luminescence, and the spectra are shown in Figure 9. The luminescence intensity decreases as the temperature increases due to the annihilation of optical centers with the rising temperature. The position of peaks does not have any shift, which indicates the position of the two Gaussian divided peaks, 543 nm (2.28 eV) and 626 nm (1.98 eV), stay stable. The behavior of 4H-SiC is different from 6H-SiC, the highest luminescence intensity of which appears at 150 K [11], and the position of peaks is different at different temperatures.

### 3.2. First-Principles Calculation

To further understand the D_I_ optical center, the origin of D_I_ was investigated, and Si_c_(C_si_)_2_ is mostly considered the structure. The intrinsic defects in SiC, such as vacancies, antisites, and interstitials, have been widely studied by first-principles calculations [28,29,30,31,32]. The optical transition energy of the Si_c_(C_si_)_2_ defect in 4H-SiC was calculated in two ways using Material Studio (MS). The difference between the two methods is the calculation of the excited state energy. First, we calculate the band structure at different charges to calculate the formation energy. The formation energy can be calculated with Equation (1).
E*_f_* [(Si_c_(C_si_)_2_)^q^] = E_tot_[(Si_C_(C_Si_)_2_)^q^] − E_tot_[(SiC)^q^] − ∑*n*_*i*_μ_i_ + q(E_F_ + ε_vbm_ +Δ*V*) + E_corr_(1)
where E*_f_* [(Si_c_(C_si_)_2_)^q^] represents the formation energy when the Si_c_(C_si_)_2_ defect is in the state of charge q. E_tot_[(Si_C_(C_Si_)_2_)^q^] is the total energy of the crystal containing the defect of Si_C_(C_Si_)_2_ at charge state q in the supercell, and E_tot_[(SiC)^q^] is the total energy of a perfect 4H-SiC crystal in the same supercell. *n*_*i*_ represents the energy change caused by adding (*n*_*i*_ > 0) or removing (*n*_*i*_ < 0) atoms in the defect system relative to the defect-free system, and *μ*_*i*_ is the chemical potential corresponding to the specific atom. E_F_ represents the Fermi energy. ε_vbm_ is the energy at the top of the valence band, and Δ*V* is the valence band top correction value of the defective system and the nondefective system. E_corr_ is the system correction needed for the calculation.

Due to the calculated formation energy, we can obtain the optical transition energy, which can be described in Equation (2):E_pl_ [q1, q2] = E_g_ − (E*_f_* [(Si_c_(C_si_)_2_)^q1^] − E*_f_* [(Si_c_(C_si_)_2_)^q2^]) − E_rel_, (2)

In the calculation of the Si_c_(C_si_)_2_ defect, *n*_*i*_ of C atoms is increased by 1, and *n*_*i*_ of Si atoms is reduced by 1. The difference between the total energy of q = 1 and q = 0 is:

ε(0/+) = E*_f_* [(Si_C_(C_Si_)_2_)^0^] − E*_f_* [(Si_C_(C_Si_)_2_)^+^] = 0.59 eV. We also calculate the formation energy when q = −1 and q = 2, and based on these results, we show the spectrum of the Si_C_(C_Si_)_2_ defect formation energy at charges of 0, 1, −1, and 2 in Figure 10. Only ε (0/+) lies in the band gap. The relaxation energy E_rel_ is 0.08 eV in our simulation, and the Fermi level was calibrated. The real impurity level is shown in Figure 11, and −0.18 eV is the real impurity energy level at the charge state of 0. The band gap of pure 4H-SiC measured in the experiment is 3.26 eV. Finally, we deduce that the optical transition energy E_pl_ is 2.77 eV (447 nm). We show the optical transition luminescence from q = 0 to q = 1 in Figure 12, which has a slight deviation from the experimental value of 2.901 eV (427 nm) using the PL measurement. The error rate is 4.5%. The difference between the first-principles calculation and experimental results means that there are unknown nonradiative composite channels for the defect Si_C_(C_Si_)_2_ and the system error of the calculation method. Thus, the deviation is regarded as reasonable, and Si_C_(C_Si_)_2_ is the composite of the D_I_ luminescence center.

Second, time-dependent density functional theory (TD-DFT) in CASTEP was used to calculate the excitation energy directly. The calculated value of pure 4H-SiC with the PBE method in CASTEP is 2.24 eV. Compared with other 4H-SiC simulations using the PBE method [33], the calculation results are reliable. The excitation energy of defect Si_c_(C_si_)_2_ is 1.9 eV, the value of E_pl_ is 2.87 eV, and the error rate of this method is 1%.

These two calculations are all based on the first principles performed by CASTEP, and the error rate of these two calculations is 4.5% and 1%, respectively. To take into account the systematic error of the calculation, the results are regarded as reasonable. We notice the defect Si_C_ from q = 1 to q = 2 was proposed as the composite of D_I_ defect [34], but we also notice there are some debates on the structure. The recombination of carbon vacancy(V_C_) and Si interstitial (Si_i_) above 1200 °C is suggested to create Si_C_ defect [35]. The results of the IBIL experiments in this paper show the D_I_ defect shows up after 900 °C annealing and is inexistent for the untreated sample. Our calculation results and experiments support the view that Si_c_(C_si_)_2_ is the structure of the D_I_ defect. To combine the two calculated results, we believe that the optical transition of the defect Si_c_(C_si_)_2_ from q = 0 to q = 1 can explain the D_I_ luminescence.

## 4. Conclusions

In conclusion, The IBIL measurements with 2 MeV H^+^ at 100 K were carried out. A wide peak lies at 597 nm, which is due to deep donor and shallow acceptor recombination, and a small peak between 420 and 430 nm was found in the annealed sample at 900 °C in air and disappeared in the untreated sample, which is attributed to the improvement in the concentration of the D_I_ optical center after annealing. The D_I_ peak and the wide peak at 597 nm drop with the accumulation of H^+^. The PBE function of DFT based on first-principles calculations is used to investigate the origin of D_I_ defects. Photon emission during the electron transfer transition from q = 0 to q = 1 of the complex defect Si_C_(C_Si_)_2_ (2.901 eV) may explain the D_I_ luminescence peak. The calculation results are in excellent agreement with the experimental values.

## Figures and Tables

**Figure 1 materials-16-02935-f001:**
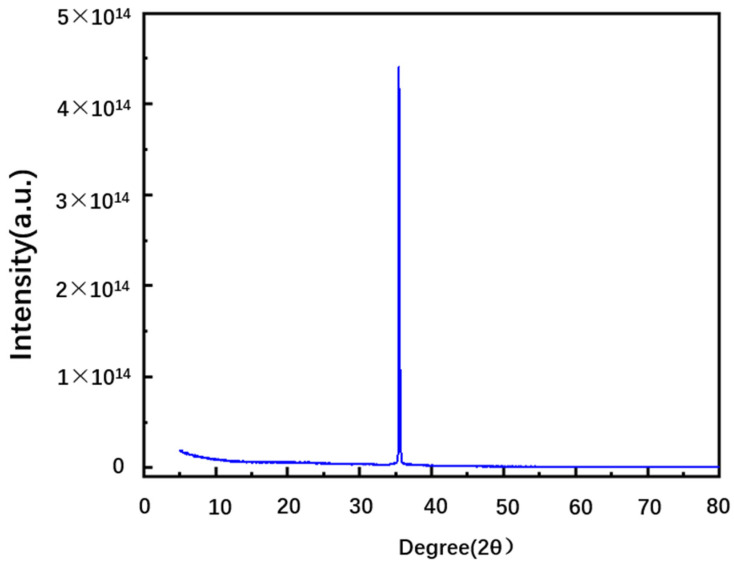
XRD measurement of the pure 4H-SiC sample.

**Figure 2 materials-16-02935-f002:**
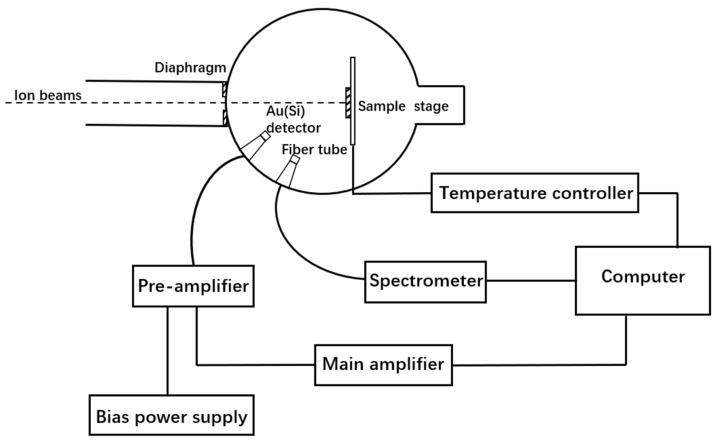
Schematic diagram of the IBIL experimental setup.

**Figure 3 materials-16-02935-f003:**
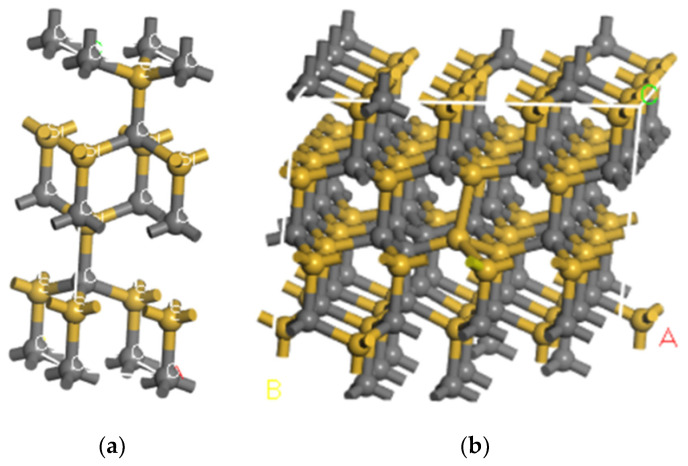
(**a**) Single cell of 4H-SiC; (**b**) 4H-SiC cell is expanded to 4 × 4 × 1 cells. (A, B and C mean the magnification times along the corresponding direction).

**Figure 4 materials-16-02935-f004:**
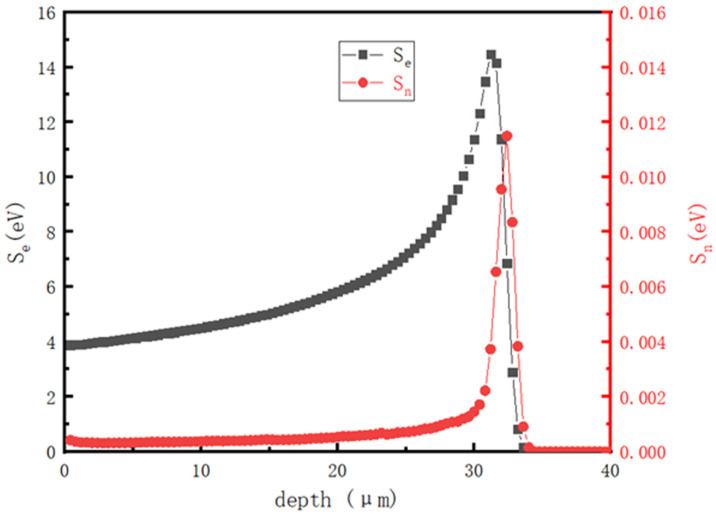
Nuclear energy loss and electron energy loss with SRIM calculation of 2 MeV H^+^ inducing 4H-SiC.

**Figure 5 materials-16-02935-f005:**
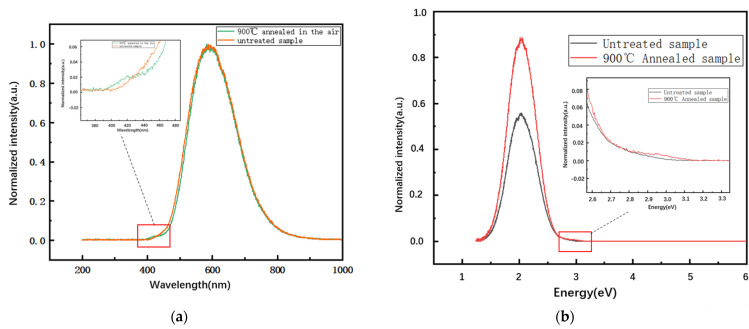
Comparison of measurement results of untreated samples and samples annealed at 900 °C with the same fluence of 1.45 × 10^11^ ions/cm^2^ at 100 K. (**a**) Wavelength spectrum; (**b**) energy spectrum.

**Figure 6 materials-16-02935-f006:**
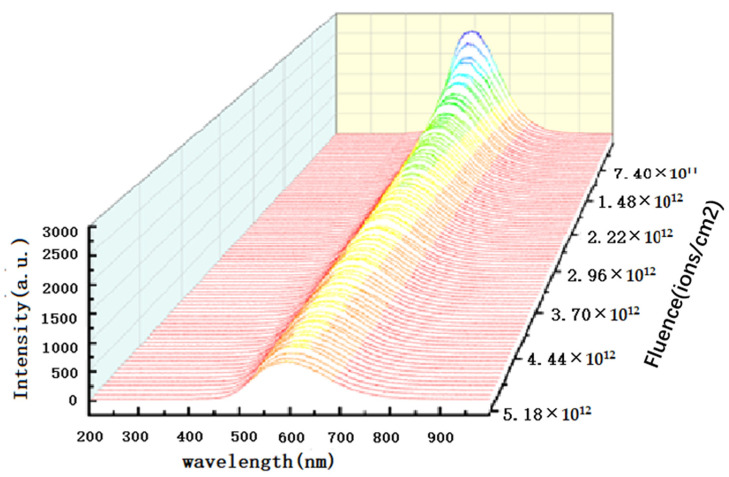
IBIL intensity decreases with ion fluence from 1.14 × 10^11^ to 5.18 × 10^12^ ions/cm^2^.

**Figure 7 materials-16-02935-f007:**
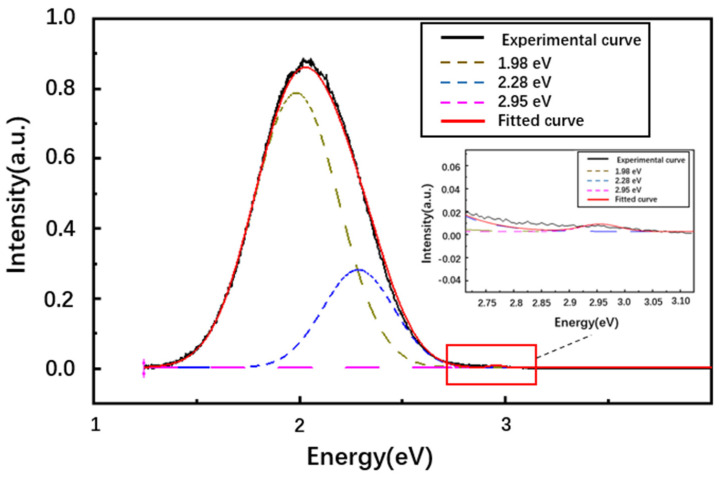
A typical Gaussian decomposition of the IBIL spectrum of the annealed 4H-SiC sample at a fluence of 1.45 × 10^11^ ions/cm^2^.

**Figure 8 materials-16-02935-f008:**
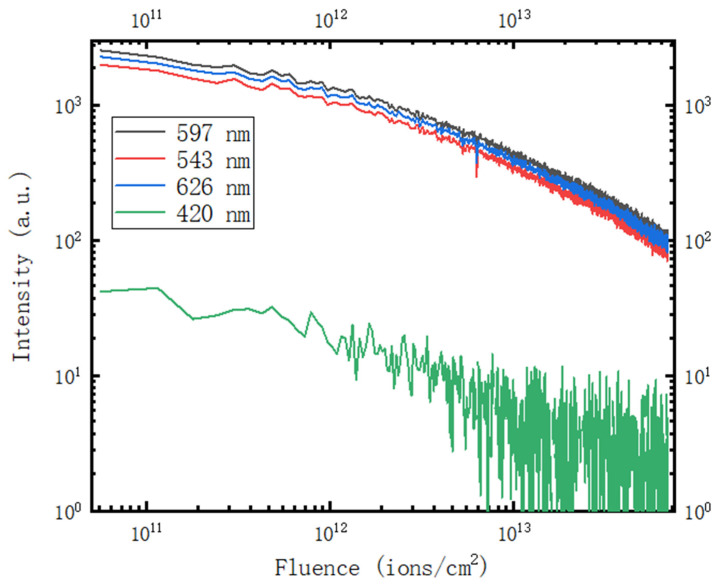
Fluence evolution of the wide band intensity at 420 nm, 597 nm, 543 nm, and 626 nm.

**Figure 9 materials-16-02935-f009:**
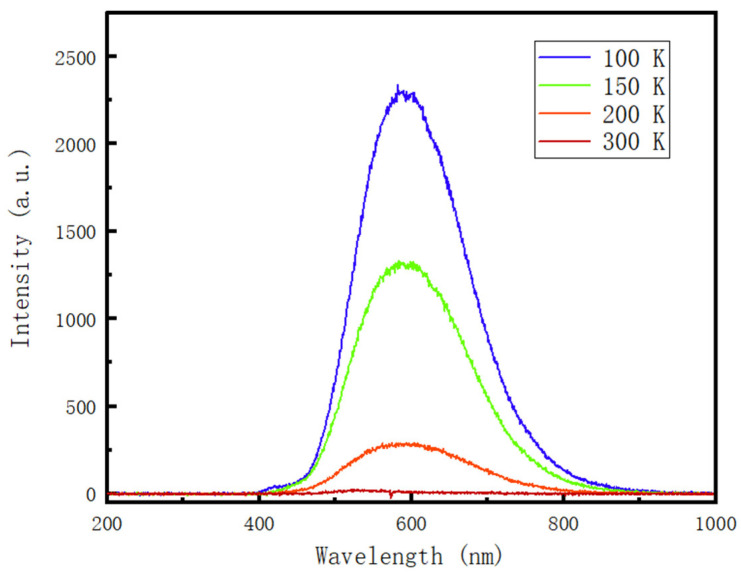
Measurement results of samples annealed at 900 °C at the fluence of 1.45 × 10^11^ ions/cm^2^ at different temperatures.

**Figure 10 materials-16-02935-f010:**
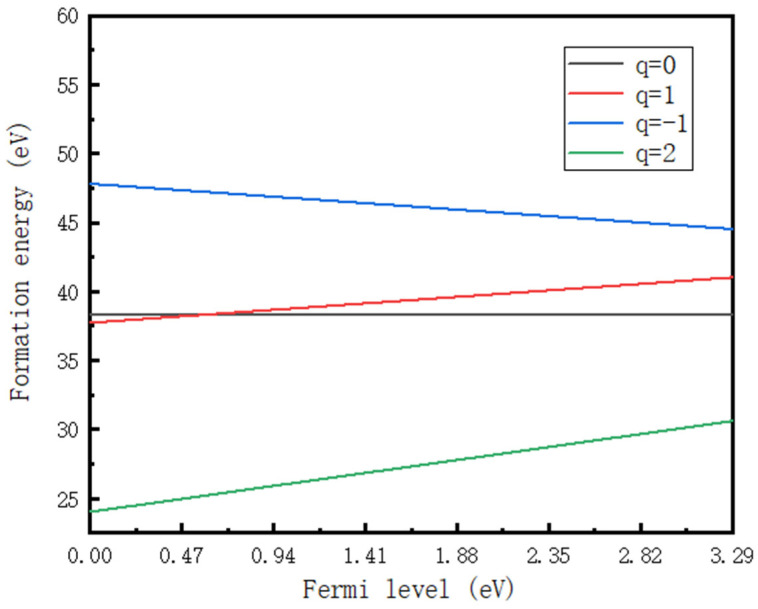
Formation energy for Si_c_(C_si_)_2_ in 4H-SiC.

**Figure 11 materials-16-02935-f011:**
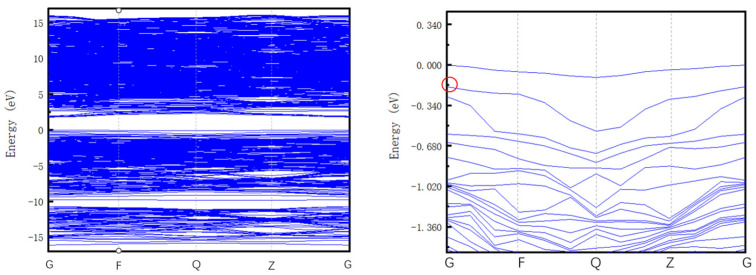
Fermi level correction of Si_c_(C_si_)_2_ defect at the charge state of 0. (The real impurity level is marked in the red circle).

**Figure 12 materials-16-02935-f012:**
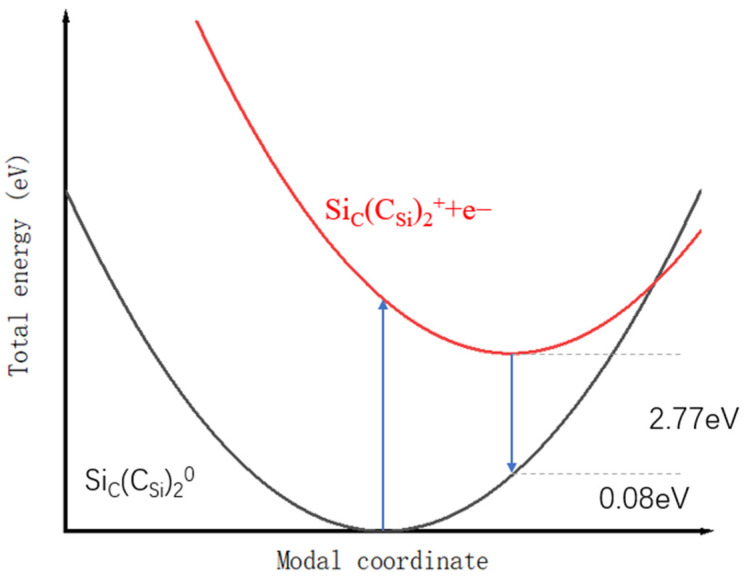
Configuration coordinate diagram for the Si_c_(C_si_)_2_ defect.

## Data Availability

The datasets generated during and/or analyzed during the current study are available from the corresponding author on reasonable request.

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
