# Peer review of "IBIL Measurement and Optical Simulation of the DI Center in 4H-SiC"

_materials, 2023, doi:10.3390/ma16072935_

Round 1
Reviewer 1 Report
1.-All acronyms (IBIL, Di, etc) must be explained the first time they appear.
2.- In figure 1, the ordinate must be arbitrary units
3.- The band gap of the sample used must be determined experimentally.
4.- The ICSD reference used must be added to ensure that in XRD that the analyzed compound is the one mentioned, in addition to establishing the corresponding space group
5.- Does the spectrum analyzed correspond to the sample before or after the heat treatment process?
6.- Was the density of the sample measured experimentally? As? If not, how do you ensure that the value used is correct?
7.- In figure 3, Sn and Se, n and e must be subscripts
8.- In Figure 4, the decrease in luminescence as a function of energy can be analyzed as a function of what type? Please graph the relationship of the luminescent decay as a function of the applied energy, and explain the phenomenon.
9.- On what basis do the authors affirm that the small emission band observed at 420 nm corresponds to the aforementioned defect?
10.- Assigning the maximum of such emission to be at 420 nm is unconvincing, please provide better results (probably doing spectrum deconvolution is convenient)
11.- If there really is the appearance of the mentioned defects after the heat treatment, why do it only after 20 min? It is evident that if it were taken to much longer times, for example, to 5 h, they would increase and the emission mentioned at 420 would be more evident, which at the moment is not very convincing.
12.- At 150 , is it possible to observe the 420 nm emission?
13.- The mechanism mentioned in lines 258-260 must be explained in greater detail
Author Response
We feel so grateful for your comments on my work, and the reply has been uploaded in the attachment.

Reviewer 2 Report
The manuscript offers an interesting version of the nature and properties of the Di center in 4H-SiC, based both on the identification of a low-intensity peak near 420 nm in the spectra, and on ab-initio calculations. At the same time, the experimental confirmation of this version is not convincing enough in the manuscript.
1. There is no citation in the list of references [Bin Chen, et al, Electron-beam-induced current and cathodoluminescence study of dislocation arrays in 4H-SiC homoepitaxial layers // JOURNAL OF APPLIED PHYSICS 106, 074502, 2009], in which “Cathodoluminescence results demonstrate that a new peak 417 nm appears at the formed rhombic stacking faults, which is the same as the phenomenon observed from dissociating basal plane dislocations". I believe that this result should be discussed in the manuscript under review.
2. The explanation of the asymmetry of the spectra (lines 160-161 in manuscript) needs to be confirmed. In Fig. 9, on the long-wavelength side of the peak (measurements at 100 and 150 K), nonmonotonicity is clearly visible near about 640 nm. It is possible that this is an instrumental effect, but it may also be a separate component in the spectrum.
3. It is necessary to describe the measurement conditions. References to one's own work [21] are not enough. From the text of the manuscript it is not clear with what spectral and spatial resolution the measurements were carried out.
4. It is necessary to illustrate how the intensity of the band with a maximum near 420 nm was determined. What is the FMHM of this band, does it depend on flux?
5. Figure 7 contains all the information that is presented in figures 5 and 6. Please justify the need for figures 5 and 6 in the manuscript.
6. Figure 8 needs to be changed. It is difficult to compare the dependences of the intensities of the two peaks on flux. What do the negative intensity values in Figure 8, b mean? What does the expression "wide band intensity at 420 nm" mean? In my opinion, the presentation of the results in one figure with a double logarithmic scale would be more visual.
7. It is necessary to give explanations on the temperature dependences of the spectra. Please explain at what temperatures the measurements were taken, what mechanisms are associated with thermal quenching of luminescence, how the luminescence intensity was calibrated, what is the thermal activation energy of luminescence quenching, whether the shape and position of the luminescence band maximum change depending on temperature. Figure 9 also needs to be corrected. It would be more clear to present the spectra as normalized or multiplied by the corresponding coefficients, and show the temperature dependence of their intensities in the box.
Author Response

(The authors gave the same response as above.)

Reviewer 3 Report
The manuscript concerns the nature of the DI center, one of the most important defects in 4H-SiC. SiC itself attracts much of interest due to its outstanding physical properties and important electronic applications. So, its characteristics have been studied with use of various experimental techniques. Features of DI centers were revealed by photoluminescence, DLTS, EPR etc. Many papers report on the attempts to describe the nature of the DI center by the first-principle calculations. Nevertheless, the specific origin and the atomic structure of the DI center are still under debate. The authors contribute to this rather timely subject with ion-beam-induced luminescence (IBIL) data and the calculation of the characteristics (optical transition energy, in particular) of a SiC(CSi)2 complex which could be identified with DI. The analysis of the IBIL spectra confirms the transition energy obtained previously with other experimental techniques. The proposed model expands the family of centers associated with DI (antisites, antisite pairs, divacancies etc.). So, the manuscript deals with an important and timely subject, matching the scope of the journal, and gives some new experimental results - the same transition energy as reported in literature but acquired with another experimental technique. The proposed model of the DI center leads to the transition energy consistent with the experimental data but it is not thoroughly compared with the models proposed by other groups, in order to show its advantages and disadvantages (for example, the recent paper by H.-S. Zhang et al. Phys.Status Solidi RRL 2023, 17, 2200239 is not even mentioned in the text). Therefore, I believe that the manuscript has to be expanded by a thorough discussion of the results, showing their strength, comparing them directly with the corresponding results available in literature. That would possibly show the importance of the authors contribution to the field and rise the scientific quality of the manuscript to the standards of the “Materials” journal.
Besides, adding some information about the samples (like the size and its crystallographic orientation) and about the XRD experiment (a set-up and conditions) would be useful .
Some other faults should be fixed:
- The unnecessary duplication of “…it is still uncertain.” In lines 58 and 59 can be removed.
- A non-English character in the line 113 should be removed.
- The curves in Figs. 5 and 6 are duplicated in Fig. 7. Do they contain any important information that cannot be extracted from Fig. 7?
- In the caption of Fig. 7, the flux is 1.5x 10^11 or 1.45x10^11?
- In the caption of Fig. 9, a capital M at the beginning is lacking and two different sizes of the font are used.
- Fig. 11 is mentioned in the text before Fig. 10.
Author Response

(The authors gave the same response as above.)

Reviewer 4 Report
This manuscript reports on the evolution of DI defects in 4H-SiC due to 2 MeV H+ implantation. The topic is technologically relevant, and the reported work and results are logically complementary. In this context, studies related to the identification of point defects and their clusters by in situ ion-luminescence spectroscopy are important and encouraged.
However, the first-principles calculations were performed on the 4H-SiC in the past extensively and DI defects formation was studied by DFT. The authors have not considered the recent reports including DOI: 10.1002/pssr.202200239, https://doi.org/10.1063/5.0051328, DOI 10.1088/1361-6463/ac3a49.
Novelty must be clearly demonstrated and reported results must be discussed in comparison with the literature. Reporting incremental results is discouraged.
Why were the experiments conducted at 100 K but not at a lower temperature?
Figs.5 and 6 are redundant because Fig.7 shows the same with greater details. Remove Figs.5&6.
Fig.10 is too busy, prepare a new figure.
Finally, it is known that SiC and SiC-based composites have a relatively small neutron displacement cross-section. Was the competing effects of ionization and nuclear energy loss in 4H-SiC from He+ ion irradiations considered and analyzed? And if not why?
The manuscript requires moderate revisions before being accepted for publication.
Author Response

(The authors gave the same response as above.)

Round 2
Reviewer 1 Report
Previous comments have been resolved
Author Response
We feel grateful for your comments on this work. The questions are answered in the attachment. Some Engllish expressions have been revised.
Reviewer 2 Report
I am satisfied with the responses to the comments and the changes made to the text of the manuscript. The authors of the manuscript have done a good job and the text has improved significantly. There are some minor remarks to the second version of the article. They refer to the decomposition of the IBIL spectrum in Fig. 7 into three components. It is clear that such a decomposition is very approximate due to the asymmetrical distribution of the vacancies of carbon and its extended defects. It would be desirable to indicate the FWHM for two more intense Gaussian contours, to agree on the positions of the maxima expressed in nm and eV (660 nm = 1.878 eV, not 1.87 eV as now indicated in the text). I note that in Figure 7 the maximum of the magenta curve is shifted from 660 nm to the long wavelength side. Judging by the appearance of the spectra in the inset to Fig. 7, the accuracy of determining FWHM is not high, therefore, it is unreasonable to indicate the FWHM values in Table 1 with an accuracy of tenths and hundredths of a nm. If the authors are confident in such an accuracy in determining the parameters of the 420 nm band, then in this case the nonmonotonicity of the change in the FWHM value depending on Fluence requires an explanation. It would be easier to remove Table 1 and write in the text, for example, that "the position and FWHM of the 420 nm band did not depend on fluence (in the range from *** to ***) and were 419-420 and 17-18 nm, respectively." The caption in figure 8 must be aligned with the notation in the box in the same figure.
Author Response
We feel grateful for your comments on this work. The questions are answered in the attachment.

Reviewer 3 Report
I feel that my comments have been seriously considered and implemented. So, I believe that the manuscript is suitable for publication, provided that a few technical faults are fixed:
- * In fig. 7 – the line types should be coherent in the graphs and in the legends, e.g. that of 660 nm maximum: broken line in the main graph but dotted line in the legend, etc.
- * The caption of Fig. 8 and the text mention only two curves (at 420 and 597 nm) while the graph contains four of them.
- * The formal ref. no. 36 is Phys. Status Solidi RRL 2023, 17, 2200239 (as taken from the journal).
Author Response

(The authors gave the same response as above.)
